# Ontology Learning Applications of Knowledge Base Construction for Microelectronic Systems Information

Frank Wawrzik *, Khushnood Adil Rafique , Farin Rahman and Christoph Grimm *

WG Design of Cyber-Physical System, TU Kaiserslautern, 67663 Kaiserslautern, Germany
* Correspondence: wawrzik@cs.uni-kl.de (F.W.); grimm@cs.uni-kl.de (C.G.)

**Abstract:** Knowledge base construction (KBC) using AI has been one of the key goals of this highly popular technology since its emergence, as it helps to comprehend everything, including relations, around us. The construction of knowledge bases can summarize a piece of text in a machine-processable and understandable way. This can prove to be valuable and assistive to knowledge engineers. In this paper, we present the application of natural language processing in the construction of knowledge bases. We demonstrate how a trained bidirectional long short-term memory or bi-LSTM neural network model can be used to construct knowledge bases in accordance with the exact ISO26262 definitions as defined in the GENIAL! Basic Ontology. We provide the system with an electronic text document from the microelectronics domain and the system attempts to create a knowledge base from the available information in textual format. This information is then expressed in the form of graphs when queried by the user. This method of information retrieval presents the user with a much more technical and comprehensive understanding of an expert piece of text. This is achieved by applying the process of named entity recognition (NER) for knowledge extraction. This paper provides a result report of the current status of our knowledge construction process and knowledge base content, as well as describes our challenges and experiences.

**Keywords:** ontology learning; ISO26262; natural language processing; named entity recognition; ontology; knowledge graph generation; classification; lessons learned; POS tagging

## 1. Introduction

Ontology learning is a specific part of automatic knoweldge base construction which involves machine learning and TBox axioms. Tbox axioms, or terminological axioms, describe information regarding the description of ideas and concepts rather than actual tangible and concrete information. It is distinguished from ontology population, which addresses the concrete instances and triples of the construction process, also called ABox or assertions box. Ontology learning includes, but is not limited to, axioms such as existential restrictions, universal restrictions, cardinality restrictions, disjunct classes, building taxonomic and subclass of hierarchies, and more. The authors of [1,2] present an overview and state-of-the-art summary.

Knowledge bases are essentially a repository of structured and unstructured information wherein they represent facts of a particular domain and demonstrate how to form logical constructs with those facts. In this paper, we focus on knowledge bases in the automotive and microelectronic domains. The manual construction of knowledge bases can be a tedious affair; therefore, we try to automate the process with as little human intervention as possible. We use a bidirectional LSTM neural network to create knowledge bases from expert articles, manuals, and just about any text from the automotive and microelectronic domains. Named entity recognition (NER) involves categorizing, identifying, and extracting named entities (or essential information) in text. In recent years, there have been many applications of NER in various domains for the classification and extraction of key information; yet, we could not find any work that extracts knowledge bases from text in the

domain of microelectronics. In this work, we have not only focused on extracting named entities but have also attempted to establish relationships between identified samples.

Background. This study is situated in the context of the German GENIAL! project. Here, we use the GENIAL! Basic Ontology (in the following, GBO) to serve as a common vocabulary for the exchange of microelectronic systems, components, functions, properties, and dependencies. We created an expressive ontology, along with an ontology suite to support the communication along the value chain through OEM to Tier 2 [3]. This approach is also coupled with a variety of tools, e.g., for performance analysis, front–end interaction, and a new approach combining ontology, documentation, and constraint analysis [4]. In this ecosystem, the ontology facilitates the usage of each term as it is defined in order to improve data quality. Its reasoning approach and results are described in more detail in [5]. The paper for the reasoning evaluation challenge made it clear that reasoning performance is still an issue, but can already be tackled. For example, an efficient reasoner such as konclude [6] with highly parallelized processing power and significant amounts of memory can be used. This, then, opens up the possibility for a first combination of reasoning and larger amounts of data, which motivated the work of this paper.

Context and contribution. Interactions with common data models have been explored [7–9] and are one way to generate knowledge out of existing data. An ontology is less useful without the knowledge it describes and thus it needs to be enhanced with an approach that facilitates automatic data aggregation with a satisfactory amount of precision. Thus, transforming linked data into information (with an ontological approach). Hence, the contribution of this paper is that we provide our experiences and results of a Bi-LSTM constructed knowledge graph of the microelectronics domain with the axiomatized expressive distinctions of the GBO ontology. We argue that this approach, namely "letting the axioms in the ontology define how the machine classifies", can potentially be useful in generating actual knowledge/information more precisely in an automatic way, which machine learning in and of itself is lacking. To the best of our knowledge, no such approach exists with respect to our domain.

This article is structured as follows: Section 2 outlines the current state-of-the-art and related work in the field of artificial intelligence in regards to knowledge base construction. Here, we also describe distinctions to our work. In Section 3, we discuss the system used for the identification and extraction of named entities in text from the automotive and microelectronic domains. Furthermore, we introduce the ontology that was used for classification. Section 4 presents the results of the system and its viability in the real world. Section 5 describes our experiences and the lessons learned. In Section 6, we conclude and discuss planned methods and tasks for future work.

## 2. Related Work

Previous authors, in their work, have either collected their own datasets [10,11] or have utilized publicly available datasets such as Genia, Cucerzan's ground truth dataset, CoNll, etc., [12,13] for their experiments. One of the primary and challenging parts of this work was to create our own dataset based on vocabulary ISO 26262. This involved carefully selecting text from relevant domains and tagging each word in its context. Drissi et al. in [14] follow a diametrical approach by constructing the ontology out of a corpus of financial data, whereas we first created the ontology with its definitions as a reference and then committed to the ontology as the basis for our descriptions. Both approaches seem valuable and complementary, depending on context and application. Loster, in their dissertation [15], recently extensively explored the topic with a focus on duplicate recognition and validation of knowledge. A way to extract relationships is, for example, through REBEL [16]. Methods such as KG-BERT [17] that are not based on foundations of ontology engineering have drawbacks in terms of understandability and consistency. Elnagar et. al. in [18] followed the direction of trying to achieve an domain-independent approach of constructing domain ontologies. Our approach is more definition-centric and expressive; however, they have more components in and support for the overall construction process. Since our background

was in building ontologies, rather than knowledge graphs, this paper also brought up the ambiguities following our work. Table 1 gives an overview of the particular distinctions from the named paper for reference. In the table, we assume that the authors made a mistake by attributing the closed-world assumption to ontologies in table row 1 (as substantiated in, e.g., [19,20]) and mistakenly swapped the easy data integration of the ontologies with the harder-to-integrate knowledge graphs (row 11, referred to in, e.g., [21,22]).

**Table 1.** Knowledge graph vs. ontology [18].

| Criteria | KG | Ontology |
|---|---|---|
| Assumption | OWA | CWA |
| Size | Massive | Relatively small |
| Scalability | Very scalable | Limited scalability |
| Scope | Problem-specific | Domain-specific |
| Real-time | Generated at runtime | Limited real-time capability |
| Timeliness | Fresh | Outdated |
| Generation | Automatic | Mostly by humans |
| Trustworthiness | Not very trustworthy | Trustworthy |
| Knowledge base type | More A-Box than T-Box | Usually more T-Box than A-Box |
| Markup language | RDF | RDF, OWL, OIL |
| Data Integration | Easily integrated | Hard to integrate |
| Quality (Correctness, Completeness) | Questionable | High Quality |
| Agility | Dynamic | Static |
| Redundancy | Very likely | Not likely |
| Reliability | Questionable | Reliable |
| Maintenance | Challenging | Burdensome |
| Evolution | Easy | Difficult |
| Security (licensing) | Questionable | Reasonable |
| Interoperability | Low | Moderate |
| Relevancy | Low | High |
| Computational Performance | Heavy | Light |

F. Niu et al., in [23], discuss a knowledge base construction (KBC) system called "Elementary", which combines machine learning and statistical inference to construct a knowledge base. The KBC model's architecture is based on Markov logic which operates on relational data. In comparison to our work, they work with significantly more data at hand for Elementary generation. However, they do not use a proper ontological model with explicit definitions, as we do. The research conducted by A. Lamurias et al. [24] introduced a new model, namely BO-LSTM, for detecting and classifying biomedical relations in documents, which utilizes domain-specific ontology by representing each entity as its ancestor sequence in the ontology. Their work mainly shows how domain-specific ontologies can improve neural network models for biomedical relations extraction, in particular for situations wherein a limited number of annotations are available. In comparison to their work, we also generate relationships; however, their approach creates an is-A or subclass of relationships, whereas we target triples. F. Ali, S. El-Sappagh, and D. Kwak [10] collected

data from two different sources: ITS office reports and social network platforms, and proposed a novel fuzzy ontology-based semantic knowledge with Word2vec embedding model to improve the task of text classification and transportation feature extraction using the Bi-LSTM algorithm. In comparison to their approach, we aim to improve the feedback of the NER results by using OWL-DL reasoning with TBox axioms, which is ongoing work, but a significant part of the overall approach. A study carried out by D. Sanchez-Cisneros and F. A Gali [25] explained that using ontology in named entity recognition tasks could be a good choice if we can choose explicit ontology. The authors proposed an ontology-based system for identifying chemical elements in biomedical documents. In comparison, instead of rules, we use the existential and universal restrictions as well as domain and range axioms for classification along with most other additional OWL-DL axioms.

Recently, constructing knowledge graphs automatically has gained increased attention as it is a disruptive technology. Recent applications can be seen in cyber-security [26], career, upskilling and education programs [27], housing market analysis [28], healthcare [29], and open drug-centric knowledge graphs [30]. Creating a knowledge graph for microelectronics is novel. Other neighboring approaches to ours are mappings [31,32]. We combine both structured and unstructured data in our approach similar to [33].

There are transformers that are more potent for NER tasks now, e.g., Autoformer, Informer, LogTrans, Reformer, and FEDformer [34–38]. However, since we had limited hardware resources and the scale of our project was on the smaller side, we tried to keep it simple and used a Bi-LSTM model for our classification task. Our Bi-LSTM model was less computationally expensive, which made it more compatible for real-time or resource-constrained applications. Transformers, on the other hand, are more computationally expensive on longer sequences.

## 3. Materials and Methods

### 3.1. Knowledge Base Construction with the GENIAL! Basic Ontology

In this section, we discuss our approach toward constructing knowledge bases with natural language processing. We also explain our experiments and the findings in detail. First, we will introduce the GENIAL! Basic Ontology. Then, we will describe how we manually created our dataset. We also explain why we use a bidirectional LSTM model for the recognition of ontological classes based on ISO 26262 and GENIAL! Basic Ontology. After introducing the KBC dataset, we will explain the process of applying our proposed approach.

### 3.2. Approach

Compared to existing approaches, which, for example, train to accumulate places, dates, objects, etc., we in our approach try to examine whether the LSTM is also able to make more fine-grained distinctions of definitions of classes. For example, recognizing the difference between a hardware part and a hardware subpart is more nuanced than between a date and a place. Figure 1 exemplifies the approach. In the first phase, we tag text articles according to our methodology and definitions and in a second step we generate the relationships between the classes.

Figure 2 shows the overall application integrated with the knowledge acquisition approach outlined in this paper. We built knowledge bases with domain experts that gather general information in text documents (picture above) and with an executable and machine-processable part (below, i.e., ElectricalSystem isA Component). Natural language text can be written within the document with pictures, videos, etc., directly in combination with the formal model. Here we read wikipedia and pdf articles as shown with its symbols. This is performed with the SysMD notebook, which is based on Markdown. On the top toolbar, with "Analyze" it is possible to calculate and propagate parameters and constraints, with "Recommend" we interface with a recommender system that proposes alternatives and related items. Our models contain basic car parts and electric components and interface with a distributed knowledge graph in the backend. Thus, mixing hand or human-defined

and computer-defined knowledge together in a database that is refined by the knowledge engineer. One of the basic ideas is to let non-knowledge engineers contribute to the knowledge base, which is then qualitatively maintained in a feedback loop, which saves time. There is a bidirectional translation from the SysMD model (based on the SysML v2 metamodel) and the knowledge graph. Outlined here is the electrical system of passenger cars and some other models (e.g., braking system, drivetrain, safety system, etc.) outlined as packages on the left. A refinement step by the knowledge engineer is then to conduct the reclassification of the electrical system as system and the components as hardware components, or to let this be performed by the reasoner itself.

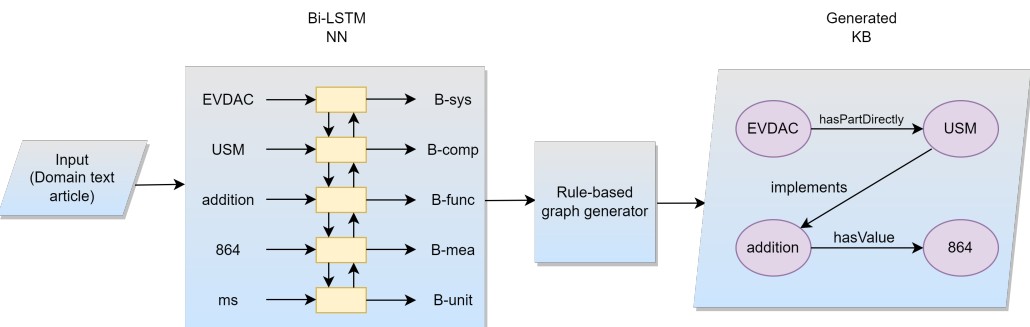

**Figure 1.** Knowledge-base construction pipeline.

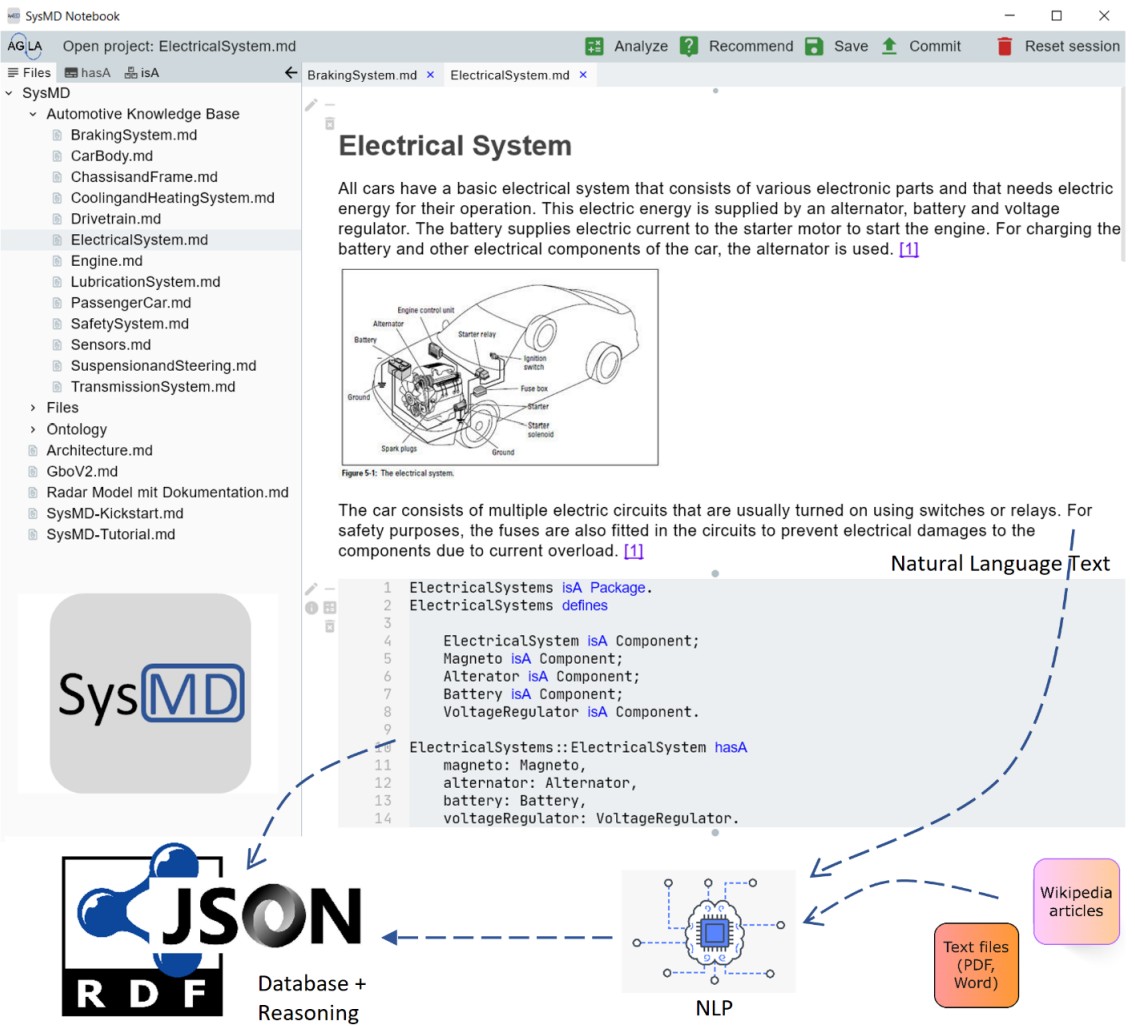

**Figure 2.** Electrical system and application.

### 3.3. Reasoning Example

Integrating the reasoner is a work in progress, with several challenges including performance on data load, and others [39]. With a simple generated relationship example, of our hardware domain ontology (no instances, and few other TBox restrictions, except those of GBO), the HermiT 1.4.3.456 reasoner took 1 min and 40 s to analyze the relationship on a Lenovo Thinkpad X1 Extreme Gen3 with core i9 and 64 GB RAM. Figure 3 shows the reasoning chain with the involved axioms, including disjoint axioms and the hierarchical "only" restrictions, which yields an erroneous classification under the owl:Nothing class for "computer". In this case, the resolution is to assign the transitive "has_part" relationship, which is a superclass of the object property "has_part_directly" as the edge/restriction.

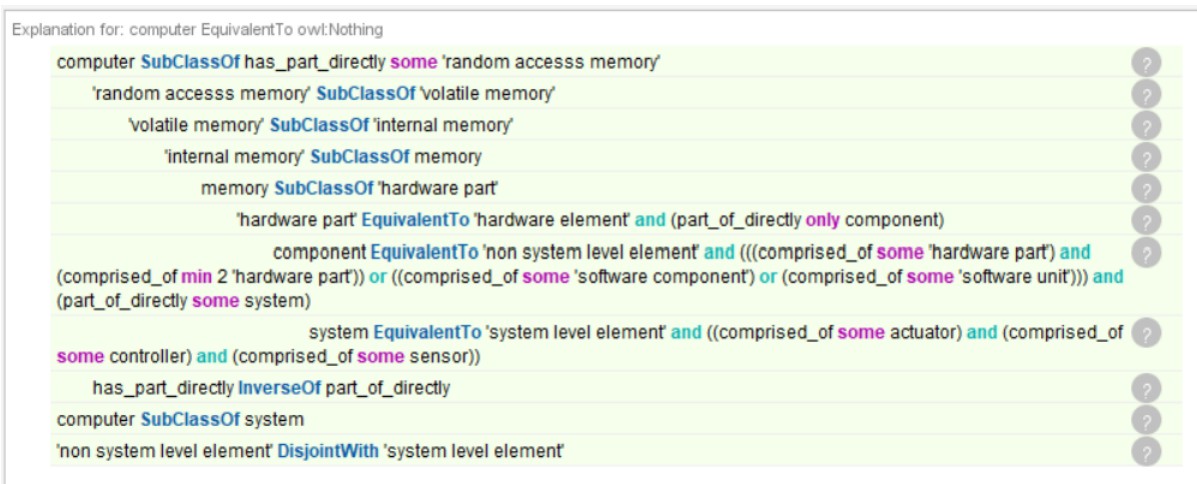

**Figure 3.** Reasoner output with NLP-generated sample relationship between computer and RAM.

### 3.4. GBO

#### 3.4.1. Overview

GBO is based on the Basic Formal Ontology [40] and ISO26262 standard in automotive electronic safety. Furthermore, better practices have been adopted and reasoning test cases created in order to validate parts of the ontology. It would be too space-consuming to print the full ontology for reference here, but we refer to our works [3] and the project repository ontologies referred to in the data availability statement. Instead, Figure 4 gives a visual outline of hardware and software with functions, so as to illustrate the use of GBO with a small example. Here, the digital filter system is classified with its parts, software, and functions. GBO's classes are shown in yellow and the hardware software domain in red. The main parts for reasoning are the universal "only" restrictions and the domain and range axioms to structure the digital filter system. The figure was created using chowlk notation https://chowlk.linkeddata.es/notation.html, (accessed on 27 November 2022). Table 2 shows the classes used for tagging with its informal as well as formal definitions.

We evaluated the ontology from various angles, including application and quality of soundness of the model. We classify it as an intermediate middle level, expressive, small (80 atomic classes), core ontology. It is not just for tagging and data aggregation, but also for use in the application itself. Several domain experts from Bosch, Infineon, Hella, etc., have given feedback and revisions. The outcome was that the domain was well represented, both in detail and correctness. Sometimes, users had preferences in terms of additions, such as including the ASIL safety level (Bosch), which was added to extensions in other parts of a module suite. This further showed that the ontology made the meaning of the words even clearer, as it was making implicit assumptions of the textual ISO26262 standard explicit [3]. Another result of the evaluation was that the model was still hard to understand and use for non-experts, and we created a more abstract simplified and intuitive vocabulary as a consequence and complement [4]. An evaluation in terms of speed took place as well [5].

The GBO ontology took a reasoning time between 2 and 15 min depending on the reasoner and machine for a hardware software domain example. The evaluation yielded necessary improvements in approaches such as modularization and allowed for the creation of this work as a next logical step.

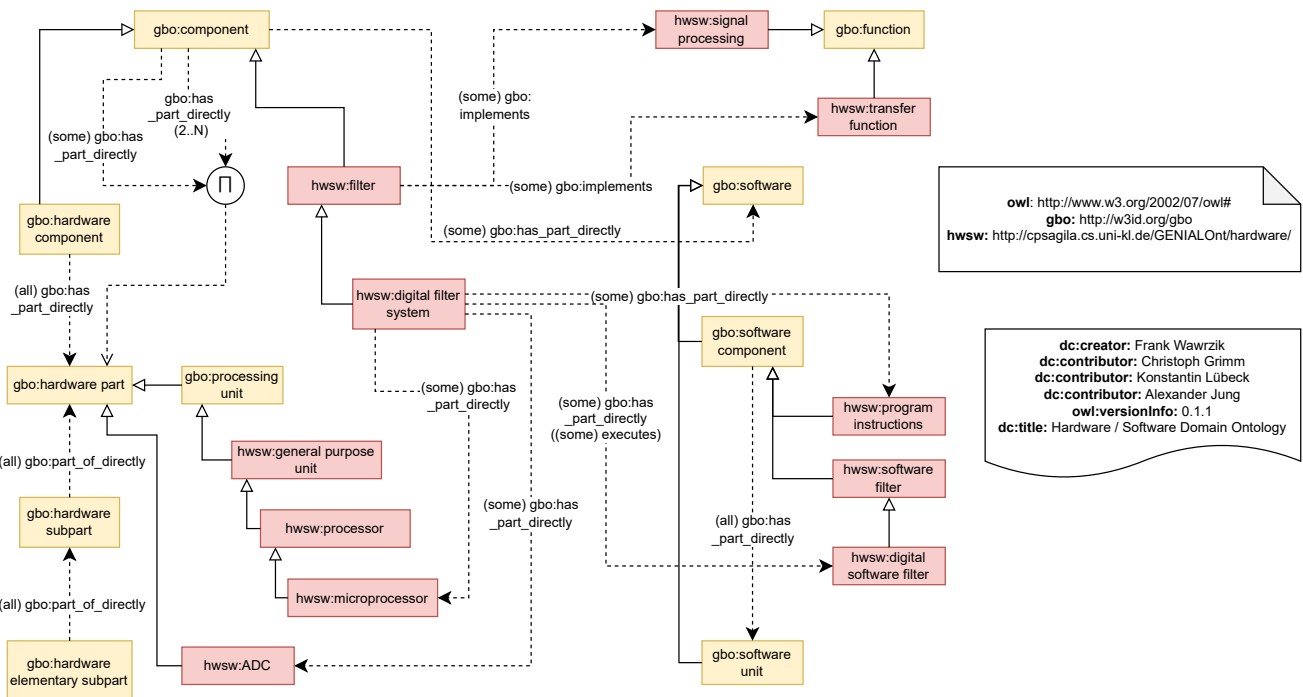

**Figure 4.** TBox reasoning of digital filter system [3].

### 3.4.2. Selecting Classes and Reducing Scope

In its current version (V 0.2.3), GBO alone contains 83 atomic classes and 14 object properties, with the expected amount of tagged classes that would require many hours of tagging for several people. In order to manage the complexity and obtain results faster, we decided to keep the amount of classes and relationships to tag to an absolute minimum, yet maintaining it useful and precise enough to be able to create a workable dataset. After a few revisions, we restrained the vocabulary from GBO to use with the following 10 classes, as can be seen in Table 2. We used a highlighting in accordance with how OWL DL is displayed in the Protégé tool. Keywords for TBox restrictions are marked in pink and logical operators connecting triples are marked in light blue. This was a compromise regarding the ideal. For example, in this set, we summarized software components and software units to software, thus having an underclassification here in comparison to the original ontology. We left out "hardware elementary subpart" because we could not find enough articles initially to cover enough tags. Furthermore, we left out "context", "mechanical object" (e.g., for acquiring car parts), "social object", and others to reduce tags. Furthermore, more precise tags such as "vehicle function" where left out.

As can be seen in Figure 4, the relationships and axioms in the ontology are designed in a more complex way in order to support more comprehensive reasoning, including inverse, symmetric, transitive, covering axioms, cardinality restrictions, and others. However, in order to generate the TBox classes and accumulation, we only use the "some" or so-called existential restrictions in OWL or normal node/edge triples for graph databases. Here, for example, the ontology suggests that all elements are related to properties with the "has property" relation. See Table 3, e.g., line 6 for the element's "has property" quantity.

**Table 2.** Tagging classes with their informal and formal definitions.

| Class | Definition |
|---|---|
| system | System level element that is according to ISO 26262: set of components (3.21) or subsystems that relates at least a sensor, a controller, and an actuator with one another.<br><br>"system level element" and ((comprised_of some actuator) and (comprised_of some controller) and (comprised_of some sensor)) |
| component | According to ISO26262: non-system level element (3.41) that is logically or technically separable and is comprised of more than one hardware part (3.71) or one or more software units (3.159).<br><br>"non system level element" and (((comprised_of some "hardware part") and (comprised_of min 2 "hardware part")) or ((comprised_of some "software component") or (comprised_of some "software unit"))) and (part_of_directly some system) |
| hardware component | According to ISO26262: non-system level element (3.41) that is logically or technically separable and is comprised of more than one hardware part (3.71).<br><br>"hardware element" and (comprised_of only "hardware part") and (comprised_of min 2 "hardware part") |
| hardware part | A piece of hardware that is (according to ISO 26262) a portion of a hardware component (3.21) at the first level of hierarchical decomposition.<br><br>"hardware element" and (part_of_directly only component) |
| hardware subpart | Portion of a hardware part (3.71) that can be logically divided and represents second or greater level of hierarchical decomposition.<br><br>"hardware element" and (has_part_directly only ("hardware elementary subpart" or "hardware subpart")) and (part_of_directly only ("hardware part" or "hardware subpart")) |
| function | A bfo:function that an element (e.g., system, component, hardware or software) implements. |
| software | From definition of element: Note 1 to entry: When "software element" or "hardware element" is used, this phrase denotes an element of software only or an element of hardware only, respectively.<br><br>"software element" is_executed_by some "processing unit" |
| quantity | A quantity is a (property that is quantifiable and a) representation of a quantifiable (standardized) aspect (such as length, mass, and time) of a phenomenon (e.g., a star, a molecule, or a food product). Quantities are classified according to similarity in their (implicit) metrological aspect, e.g., the length of my table and the length of my chair are both classified as length. |
| measure | A bfo:quality that are amounts of quantities.<br><br>hasNumericalValue some rdfs:Literal |
| unit | A quality that is any standard used for comparison in measurements. |

**Table 3.** Class relations.

| Class I (Subject) | Relation (Predicate) | Class II (Object) |
| --- | --- | --- |
| system | has part directly | component |
| hardware component | has part directly | hardware part |
| element [a] | implements | function |
| processing unit | executes | software |
| hardware subpart | part of directly | hardware part |
| element [a] | has property | quantity |
| quantity | has value | measure |
| measure | has unit | unit |

[a] comprises classes: component, hardware component, hardware part, hardware subpart, software and system.

### 3.5. Dataset

We hired two assistants who were trained with the vocabulary by the ontologist for approximately 20 h as an introduction as well as on the first hundred tags. The assistants were master students in computer science. One assistant was a male in his late twenties while the other was a female in her mid-twenties. Even though it was their second language, both could use the English language fluently and accurately in a professional setting. Furthermore, they continued to be supervised during the process for correction and questions. Tagging for 20 h a week for 3 months yielded ['B-hwp': 1426, 'B-comp': 319, 'B-hwc': 902, 'B-hwsp': 362, 'B-sw': 364, 'B-sys': 583, 'B-mea': 400, 'B-unit': 520, 'B-func': 597, 'B-qt': 1770] tags.

Figure 5 shows a direct comparison of tags of our dataset. In comparison to other datasets, our number of tags is relatively small. Table 4 gives an overview of the tagged articles and the number of all tags made for the article.

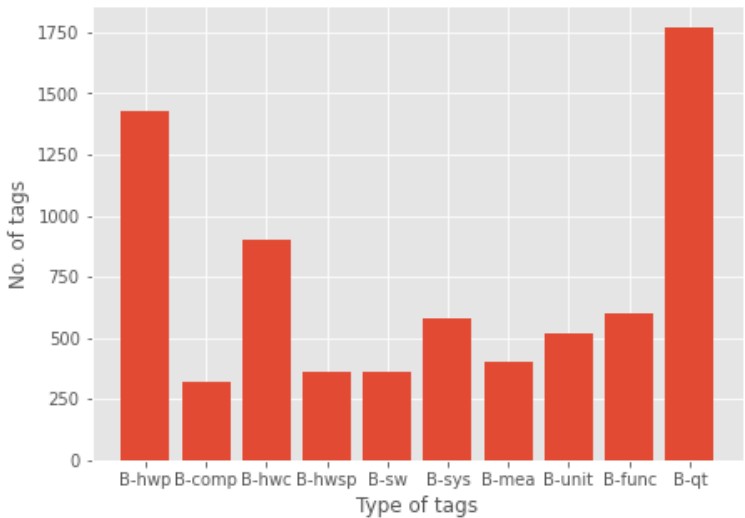

**Figure 5.** Distribution of class examples.

**Table 4.** List of Wikipedia articles and the number of labeled tags.

| No. | Article Name | Approximate Number of Tags |
|---|---|---|
| 1 | Adaptive cruise control | 270 |
| 2 | Arithmetic logic unit | 280 |
| 3 | Cache (computing) | 400 |
| 4 | Analog to digital converter | 592 |
| 5 | Charge Pump | 112 |
| 6 | Central Processing Unit | 900 |
| 7 | Digital image processing | 100 |
| 8 | Electronic filter | 200 |
| 9 | Floating-point unit | 170 |
| 10 | Hard disk drive | 1288 |
| 11 | Latency (engineering) | 180 |
| 12 | Motherboard | 390 |
| 13 | Network interface controller | 150 |
| 14 | Random-access memory | 500 |
| 15 | Software | 200 |
| 16 | Texture mapping unit | 120 |
| 17 | Voltage controlled oscillator | 231 |
| 18 | Power supply | 360 |
| 19 | Microcontroller | 800 |

*3.6. Application of Bi-LSTM*

In order to train the model, we used a bidirectional long-short term memory (bi-LSTM) network. Bidirectional neural networks are designed such that the input travels in both directions—backward and forward [41]. The hidden layers of the network that are of opposite directions are connected to the same output. This design ensures that the output layer is fed with information from the future (forward) and the past (backward).

We applied a bi-LSTM network for this particular task because LSTM networks are effective at sequence prediction problems [41]. LSTM networks are a type of recurrent neural network (RNN) that are reliable for building language models [42]. RNNs have shown good success at word-level predictions, such as named entity recognition. An RNN is capable of storing information history in an internal memory. This information lets the network predict the current output based on its knowledge of the past [33].

Figure 6 represents the architecture of the bi-LSTM model we used to train and test our dataset. For the input layer, we defined the shape of input equal to the maximum length of a sentence. Hence, our LSTM network is considered to take input sentences of a size no larger than fifty. Layer 2 or the embedding layer is initialized with random weights to learn the embeddings of all the words in the training data. The embedding layer of Keras Keras, by Chollet, Francois et al., at https://github.com/fchollet/keras, (accessed on 6 September 2022) is used for text data in any NLP task. We already converted out words and tags into index numbers so that the embedding layer can take integer encoded data as input and then represent each word by a unique integer. Layer 3 or spatial dropout1D layer takes input from the previous layer (None, 50) and produces output (None, 50, 50). This layer is used to drop the entire 1D feature maps instead of individual features. The bi-directional layer takes the output from the previous embedding layer (None, 50, 50). Layer 4 or Bi-directional LSTM layer has two hidden layers that process the input words in both directions and produce backward and forward outputs. Both outputs are connected to the next layer. Outputs are concatenated by default, which doubles the output for the next layer. The number of LSTM cells is passed as an argument in this layer which defines how dense the LSTM layer would be. We specify this model with 256 units which, in turn, doubles the number of outputs for the next layer. In our case, it becomes 512 ($256 \times 2$) for the next layer. Layer 5 or time-distributed layer (Dense) is the output layer. It receives the input dimension (None, 50, 512) from the previous layer and provides the final output, which is the maximum length and maximum number of tags. Our label count is 21 and

therefore the output of this layer is (None, 50, 21). We trained the LSTM model with index numbers since it cannot recognize text. Every word and tag was assigned a unique index number to simplify the process. Hence, the following two new rows (Word_to_index and Tag_to_index) were generated in the dataset: Word_to_index: Assign an index number to each word. These numbers were used during training to identify the words. Tag_to_index: Assign an index number to each tag. These numbers were used during training to identify the tags. We trained for 50 epochs as that gave us the best training loss curve where over- and under-fitting were best avoided. The training loss curve remained below the validation loss curve. We used "categorical crossentropy", which we found most appropriate for a multi-class classification task. Other hyper-parameters such as the LSTM units or dropouts were adjusted with multiple passes of experimentations and comparisons. As a result, we reached the validation accuracy of 94%. Figure 6 gives an overview of this.

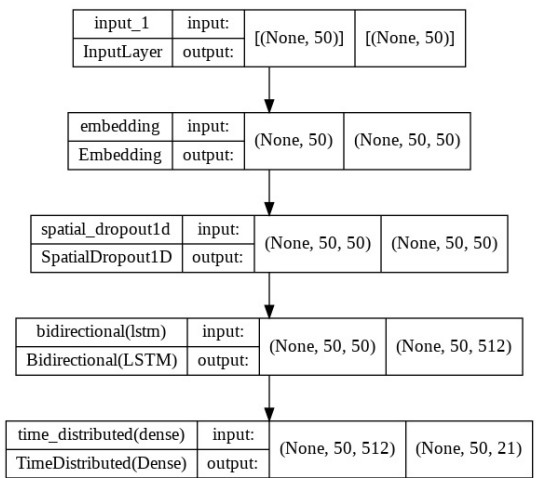

**Figure 6.** Architecture of the applied bi-directional LSTM model.

### *3.7. Data Preparation*

Named entity recognition is the task of assigning a named entity label to every word of a sentence [43]. Since we created our own dataset, we used a tagging scheme proposed by Ramshaw et al. (1999) [44] named IOB (Inside, Outside, Beginning) tagging. This format represents sentences in a way where each token is labeled "B" if the token is the start of a named entity, "I" if it is inside a named entity, and "O" indicates a token that belongs to no named entity. Punctuation marks and spaces were tagged in the same way as words, which is "O". We had a total of 2307 sentences tagged with our labels. We trained our model with 1845 sentences (80%). The training was carried out for 50 epochs with a batch size of 32. With an NVIDIA RTX 3070, it took approximately 20 min to train the model with an acceptable loss curve.

### *3.8. Relationship Establishment*

Once the model was trained and tested on validation data, we attempted to establish relationships between recognized elements. Our approach to generating knowledge bases was divided into two parts—firstly, to recognize the relevant elements in the presented text using the NER strategy, while the second step involved creating a graph using the identified elements. The elements would consequently form nodes and the edges would be constructed with the propositions presented in Table 3.

We began by defining a *context*. A *context* represented a fraction of the text or the number of successive sentences that could potentially keep the interrelated conditions intact. In our case, we found that assigning the *context* with a value within the range of 5 to 10 sentences produced the most acceptable results. Once the *context* was set, we extracted the identified words from those sentences.

We then ran a simple rule-based algorithm that contained nested if-else loops. Upon creating a context of 10 sentences, the following list 1 (one of the many lists) was produced. Figure 7 shows the graph created from the list. We chose list 1 for demonstration because it presents good diversity in terms of difference in elements.

List 1: {$system_1$, $quantity_1$, $measure_1$, $unit_1$, $component_1$, $sw\_component_1$, $function_1$, $quantity_2$, $measure_2$, $hw\_component_1$, $function_2$, $sw\_component_2$, $system_2$, $measure_3$, $unit_2$}.

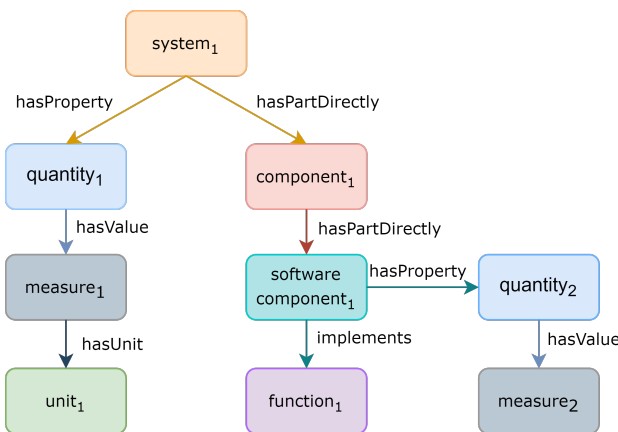

**Figure 7.** The final output with a sample context of ten sentences.

## 4. Results

When addressing NER tasks, model accuracy is not enough for evaluation due to the "O" tag or "no tag". The "no tag" labeled examples easily outnumber the samples from any other given class. The model, in almost every case, recognizes most of the "no tag" samples correctly, which consequently results in higher accuracy. Moreover, in cases of financial, medical, or legal documents, it is very important to identify precisely named entities to avoid the business cost. Therefore, to evaluate our model we calculated the F1-score [45] as well as other evaluation metrics such as precision, recall, and support on test data (refer to Table 5).

**Table 5.** Classification report.

|  | Precision | Recall | F1-Score | Support |
|---|---|---|---|---|
| comp | 0.65 | 0.63 | 0.64 | 51 |
| func | 0.35 | 0.38 | 0.36 | 98 |
| hwc | 0.66 | 0.59 | 0.62 | 193 |
| hwp | 0.65 | 0.64 | 0.64 | 307 |
| hwsp | 0.56 | 0.52 | 0.54 | 77 |
| mea | 0.68 | 0.88 | 0.77 | 72 |
| qt | 0.73 | 0.67 | 0.70 | 402 |
| sw | 0.51 | 0.58 | 0.54 | 57 |
| sys | 0.46 | 0.53 | 0.49 | 86 |
| unit | 0.75 | 0.81 | 0.78 | 116 |
| micro avg | 0.64 | 0.64 | 0.64 | 1459 |
| macro avg | 0.60 | 0.62 | 0.61 | 1459 |
| weighted avg | 0.64 | 0.64 | 0.64 | 1459 |

The precision for class "unit" was the highest at 75%, followed by "quantity" (73%) and "measure" (68%). The "function" class scored the lowest precision of 35%. Recall, on the other hand was the highest for "measure" at 88%, followed closely by "unit" (81%) and "quantity" (67%). A model can achieve high recall but incredibly low precision. In that case, we need to incorporate both the quality and the completeness of the predictions into a single score called F1-score. This measures the quality of the predictions and creates a balance between precision and recall. As mentioned in Section 5.2, there was a substantial class

imbalance and that was reflected in the F1-score being the lowest for the class "function" and the highest for the class "unit". It is the most important metric to understand how our system actually performs. Our system managed to obtain an average F1-score of 63%. Support is the number of samples on which each metric is computed. According to the classification report, precision, recall, and F1-score were calculated on 51 samples with the class label "component". Similarly, "function" was calculated on 98 samples for these metrics. The model achieved an average support of 1459, which is the sum of all supports across 10 classes.

In the case of a multi-class class imbalance dataset, the micro average metric plays an important role. If the data is perfectly balanced, the micro and macro averages will score the same result. However, if the majority class performs better than the minority ones, the micro average will be higher than the macro average, which is observable in the classification report. The macro average counts the average for precision, recall, and F1 score for an individual class. In this case, the class imbalance is not taken into consideration. Although we have a significant class imbalance, the classification report gives a closer value for precision (0.60) and recall (0.62) macro averages. A weighted average metric is also preferred for imbalanced datasets such as ours. Weighted means that each metric is calculated with respect to how many samples there were in each class. This metric favors the majority class and consequently gives a higher value when one class outperforms another due to having more samples.

From the classification report in Table 5, it is clearly observed that underclassififcation and overclassification affect system performance. Furthermore, our experiment showed that increasing the number of classes to strive for a more balanced dataset improved the overall F1-score.

Previously, we had discussed the evaluation metrics used to evaluate the model on test data that was represented by 20% of the entire dataset. Test data are data that are held back from training the model. Table 6 shows some of the predictions made by the model on test data after being trained for 50 epochs.

**Table 6.** Prediction report on test data.

| Token | True Label | Predicted Label |
|---|---|---|
| computer | B-sys | B-sys |
| , | O | O |
| RAM | B-hwc | B-hwc |
| disk | I-hwc | I-hwc |
| , | O | O |
| data | B-qt | B-qt |
| density | I-qt | I-qt |
| , | O | O |
| 109 | B-mea | B-mea |
| bit | B-unit | B-unit |
| / | I-unit | I-unit |
| s | I-unit | I-unit |
| , | O | O |
| square | B-func | B-func |
| root | I-func | I-func |
| operations | I-func | I-func |
| , | O | O |
| graphics | B-hwp | B-hwp |
| processor | I-hwp | I-hwp |
| , | O | O |
| preview | B-sys | O |
| Distance | I-sys | B-func |
| control | I-sys | I-func |
| , | O | O |
| NAND | B-hwc | B-hwp |
| drive | I-hwc | I-hwp |
| , | O | O |
| lower | O | B-func |
| frequencies | B-qt | I-func |

The table contains tokens from approximately 10–12 successive sentences. We created a CSV file of the predicted results and thoroughly reviewed it. Although all the 10 classes were covered in the test dataset, the results show that the model predicted a good number of "quantity" and "hardware part" samples, successfully. The model also performed well in predicting "hardware subpart", "measure", "unit", and "system" samples. The F1-scores clearly reflect the success of the model with these classes. Figure 8 demonstrates the knowledge graph that was created from Table 6.

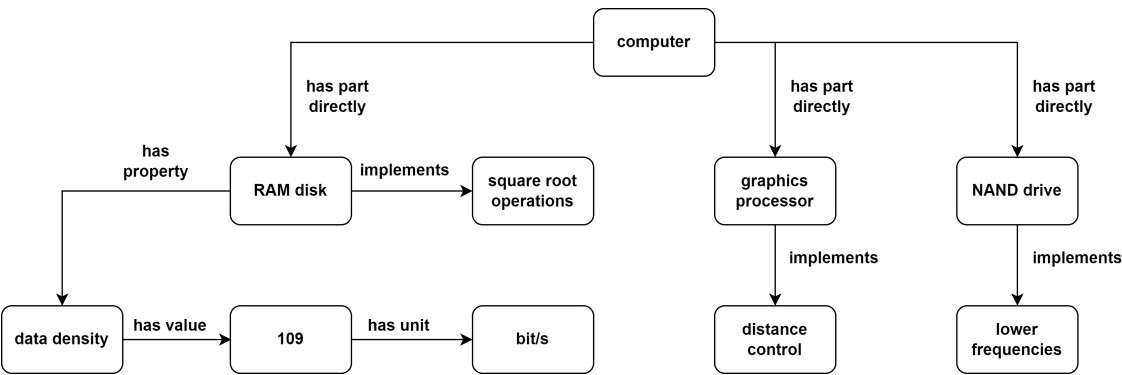

**Figure 8.** Knowledge graph generated with tokens from test predictions presented in Table 6.

The knowledge graph algorithm only takes into account the predicted examples and therefore some erroneous relationships could possibly occur. In this case, *distance control* was mislabeled as a "function", which resulted in it appearing as a function implemented by *graphics processor* (see Figure 8). The knowledge graph, therefore, can be only as accurate as the predictions made by the model and the accuracy of the expert labeling. In regard to this, Table 7 shows some predictions made that did not come from the human tagged corpus and were machine predictions.

**Table 7.** Predictions on unseen data.

| Token | Prediction |
|---|---|
| read | B-hwp |
| only | I-hwp |
| memory | I-hwp |
| , | O |
| addition | B-func |
| , | O |
| speed | B-qt |
| , | O |
| written | B-func |
| , | O |
| System/370 | B-sys |
| , | O |
| Apollo | B-sys |
| Guidance | I-sys |
| computer | I-sys |
| , | O |
| hard | B-hwc |
| disks | I-hwp |
| , | O |
| memory | B-hwp |
| cards | I-hwp |
| , | O |
| Keyboard | B-hwc |
| , | O |
| EPROM | B-unit |
| chips | I-unit |

While discussing the shortcomings of the trained model, it is important to point out that it struggled to consistently predict "function" and "software" samples, and hence they

achieved the lowest F1-scores. Some "function" samples such as *read*, *write*, *execute*, *encode*, and *decode* appeared multiple times during training and thus were predicted correctly on almost all occasions. However, other function samples such as *inverting*, *sampling*, *scaling*, *transforming*, etc., could not be predicted correctly due to an insufficient number of training examples. Table 8 illustrates some of the failed predictions.

**Table 8.** Failed predictions on test data.

| Token | True Label | Predicted Label |
|---|---|---|
| SRAM | B-hwsp | B-comp |
| caches | B-hwsp | I-comp |
| , | O | O |
| transmission | B-func | B-qt |
| , | O | O |
| lower | O | B-hwsp |
| unit | B-qt | I-hwsp |
| cost | I-qt | I-hwsp |
| , | O | O |
| Write | B-qt | I-qt |
| operation | I-qt | I-qt |
| , | O | O |
| SAS | B-comp | B-hwp |
| RAID | I-comp | B-hwp |
| Controller | I-comp | B-hwp |

## 4.1. Performance on Unseen Data

To test the model on unseen data, we selected a Wikipedia article called *"Read-only memory"* Wikipedia contributors, Read-only memory at https://en.wikipedia.org/wiki/Read-only_memory, (accessed on 12 August 2022) which is a technical article from the microelectronics domain. Table 7 demonstrates some of the predictions made by the model on the article text. The results show that the trained bi-LSTM model correctly predicted a good number of class samples. This information could then be retrieved in the form of sets of graphs similar to Figure 8. Figure 9 is a screenshot of the knowledge graph generated by our first experimentation with a transformer neural network and demonstrates the relationship generation between the entities.

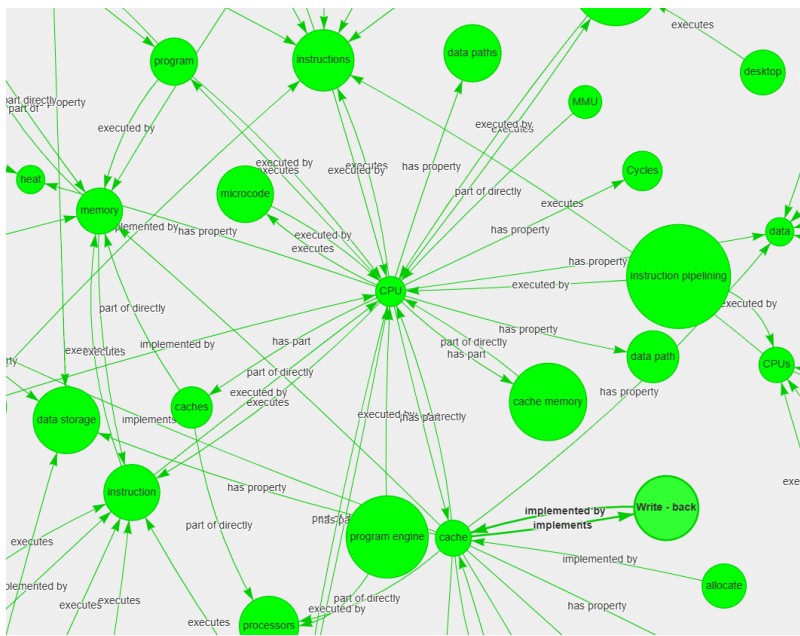

**Figure 9.** Knowledge graph generated on our test data using a transformer neural network.

## 4.2. Validation

Our validation process is two-fold, as follows.

### 4.2.1. Manually

After the graph is generated, we mostly check for semantic accuracy, coherency, and consistency manually and update the dataset. Other criteria such as timeliness, understandability, and completeness are out of our scope. Two validation examples are given as follows:

- The transformer uses a relationship "uses" between parts and functions, we classify it as subclass of "implements" in order to adhere to our schema and retain the other information as well (coherency);
- We recognize whether a property was falsely classified as a function, and readjust the dataset and graph (semantic accuracy).

### 4.2.2. Automatically

We apply the reasoner and create an inconsistent knowledge graph and dataset as well as a consistent one automatically, which simplifies manual work.

## 5. Discussion

### 5.1. Particular Examples and Their Considerations

In this subsection, we give a few examples of text sentences and POS tagging, discuss results, challenges, and implications. The sentences can be viewed in Table 9.

(1) A central processing unit (CPU), also called a central processor, main processor, or just processor, is the electronic circuitry that executes instructions comprised in a computer program.

From the ontologists' point of view, this sentence was classified correctly by the assistant. Notably, electronic circuitry was not tagged, which is correct, because it is an element, but not any one of the tagging elements. Rather, it can be classified as the superclass of integrated circuit but still as an element. Correct taxonomies are an essential current [46] and future challenge. Secondly, naturally the processor executes instructions, which validates our GBO ontology. Thirdly, the relationship is to be constructed between processor and instructions and not with the electric circuitry, which would be an overclassification and not existentially correct. Not all electronic circuits execute instructions. Further instructions comprise a computer program in the text. However, an instruction is in fact a part of a computer program with the ontology we designed. Furthermore, a computer program (software component) is comprises instructions (software unit) using the inverse relationship. This exemplifies the intricacies of natural language and how a loose usage may potentially confuse the machine.

(2) The CPU performs basic arithmetic, logic, controlling, and input/output (I/O) operations specified by the instructions in the program.

As can be seen, the IOB tagging scheme cannot be easily applied, as it is (1) arithmetic operations, (2) logic operations, and (3) controlling operations. However, as a list with commas, not allowing for an direct I-tag. The choices for tackling this problem may have different implications. We chose to just tag the first word of each compound.

(3) The principal components of a CPU include the arithmetic–logic unit (ALU) that performs arithmetic and logic operations, a processor that registers that supply operands to the ALU and stores the results of ALU operations, and a control unit that orchestrates the fetching (from memory), decoding, and execution (of instructions) by directing the coordinated operations of the ALU, registers, and other components.

This sentence shows a limitation of the simplified vocabulary, which can express that the ALU implements (orchestrates) fetching. However, not the fetching from memory. Here, we show more of the POS tagging that has been performed, to help researchers understand our tagging scheme and illustrate the examples given. Each sentence is separated by a line for the sake of readability.

**Table 9.** Central processing unit article with POS tags.

| Token | POS | Label |
|---|---|---|
| A | DET | O |
| central | ADJ | B-hwp |
| processing | NOUN | I-hwp |
| unit | NOUN | I-hwp |
| ( | PUNCT | O |
| CPU | PROPN | B-hwp |
| ) | PUNCT | O |
| 0 | PUNCT | O |
| also | ADV | O |
| called | VERB | O |
| a | DET | O |
| central | ADV | B-hwp |
| processor | NOUN | I-hwp |
| 0 | PUNCT | O |
| main | ADJ | B-hwp |
| processor | NOUN | I-hwp |
| or | CCONJ | O |
| just | ADV | O |
| processor | NOUN | B-hwp |
| 0 | PUNCT | O |
| is | AUX | O |
| the | DET | O |
| electronic | ADJ | O |
| circuitry | NOUN | O |
| that | PRON | O |
| executes | VERB | O |
| instructions | NOUN | B-sw |
| comprising | VERB | O |
| a | DET | O |
| computer | NOUN | B-sw |
| program | NOUN | I-sw |
| . | PUNCT | O |
| The | DET | O |
| CPU | NOUN | B-hwp |
| performs | VERB | O |
| basic | ADJ | O |
| arithmetic | ADJ | B-func |
| 0 | PUNCT | O |
| logic | NOUN | B-func |
| 0 | PUNCT | O |
| controlling | VERB | B-func |
| 0 | PUNCT | O |
| and | CCONJ | O |
| input | NOUN | B-func |
| / | SYM | I-func |
| output | NOUN | I-func |
| operations | NOUN | I-func |
| specified | VERB | O |
| by | ADP | O |
| the | DET | O |
| instructions | NOUN | B-sw |
| in | ADP | O |
| the | DET | O |
| program | NOUN | B-sw |
| . | PUNCT | O |
| This | PRON | O |
| contrasts | VERB | O |
| with | ADP | O |
| external | ADJ | O |
| components | NOUN | O |
| such | ADJ | O |
| as | ADV | O |
| main | ADJ | B-hwp |
| memory | NOUN | I-hwp |
| and | CCONJ | O |
| I | NOUN | B-hwp |
| / | SYM | I-hwp |
| O | NOUN | I-hwp |
| circuitry | NOUN | I-hwp |

**Table 9.** *Cont.*

| Token | POS | Label |
|---|---|---|
| 0 | PUNCT | O |
| and | CCONJ | O |
| specialized | ADJ | B-hwp |
| processors | NOUN | I-hwp |
| such | ADJ | O |
| as | ADP | O |
| graphics | NOUN | B-hwp |
| processing | NOUN | I-hwp |
| units | NOUN | I-hwp |
| ( | PUNCT | O |
| GPUs | NOUN | B-hwp |
| ) | PUNCT | O |
| . | PUNCT | O |
| The | PUNCT | O |
| form | PRON | O |
| 0 | VERB | O |
| design | NOUN | O |
| ... | ... | .. |
| Principal | ADJ | O |
| components | NOUN | O |
| of | ADP | O |
| a | DET | O |
| CPU | NOUN | B-hwp |
| include | VERB | O |
| the | DET | O |
| arithmetic | ADJ | B-hwsp |
| - | PUNCT | I-hwsp |
| logic | NOUN | I-hwsp |
| unit | NOUN | I-hwsp |
| ( | PUNCT | O |
| ALU | NOUN | B-hwsp |
| ) | PUNCT | O |
| that | PRON | O |
| performs | VERB | O |
| arithmetic | ADJ | B-func |
| and | CCONJ | O |
| logic | NOUN | B-func |
| operations | NOUN | I-func |
| 0 | PUNCT | O |
| processor | NOUN | O |
| registers | NOUN | O |
| that | NOUN | O |
| supply | NOUN | O |
| operands | VERB | B-qt |
| to | ADP | O |
| the | DET | O |
| ALU | NOUN | B-hwsp |
| and | CCONJ | O |
| store | VERB | O |
| the | DET | O |
| results | NOUN | O |
| of | ADP | O |
| ALU | ADJ | B-func |
| operations | NOUN | I-func |
| 0 | PUNCT | O |
| and | CCONJ | O |
| a | DET | O |
| control | NOUN | B-hwsp |

**Table 9.** *Cont.*

| Token | POS | Label |
|---|---|---|
| unit | NOUN | I-hwsp |
| that | PRON | O |
| orchestrates | VERB | O |
| the | DET | O |
| fetching | NOUN | B-func |
| ( | PUNCT | O |
| from | ADP | O |
| memory | NOUN | B-hwp |
| ) | PUNCT | O |
| 0 | PUNCT | O |
| decoding | VERB | B-func |
| and | CCONJ | O |
| execution | VERB | B-func |
| of | ADP | O |
| instructions | NOUN | B-sw |
| by | ADP | O |
| directing | VERB | O |
| the | DET | O |
| coordinated | VERB | O |
| operations | NOUN | O |
| of | ADP | O |
| the | DET | O |
| ALU | PROPN | B-hwsp |
| 0 | PUNCT | O |
| registers | NOUN | O |
| and | CCONJ | O |
| other | ADJ | O |
| components | NOUN | O |
| . | PUNCT | O |

## 5.2. General Challenges

The application of NER techniques to achieve our goals came with several challenges, the most severe of them being a small and imbalanced dataset. In this section, we discuss the challenges we faced and also outline challenges faced by other researchers whose works are similar to ours, and then point out some of the ways to overcome them.

### 5.2.1. Classification

- **Over and under-classification:** While labeling data, we handpicked several microelectronics articles related to our ontology classes, and while labeling about 2307 sentences, we tagged a total of 1770 "quantities", and this was a relatively high number compared to other classes, such as "component", "hardware subpart", and "software". The class "hardware part" similarly had a rather high count of 1426. These differences in class distribution led to dataset imbalances [47]. Moreover, we tagged examples of the "function" class from several different articles which were not similar to each other. Despite having a good count of "function" tags, our model initially struggled to predict the "function" class examples. It could only predict "function" examples that it had seen multiple times during training. Hence, the model achieved the lowest F1-score for the "function" class (Table 5). One of the most widely adopted techniques for overcoming imbalanced datasets is resampling data points. We used resampling to over-sample the minority classes by adding more examples. We applied the simplest implementation of over-sampling which is to duplicate random records from the minority class. However, we used oversampling sparingly to avoid the likelihood of overfitting. Figure 5, that we showed earlier, illustrates the class imbalance that we had to address. The oversampling technique alleviated the situation to an extent

but the overall results could still be improved by adding more novel examples in new contexts.

- **Adequacy of the current ontology model:** We noticed that the ontological commitment was very tight and fitting to our ontology in general. For example, we defined a covering axiom to make the description of "hardware" complete with "hardware component", "hardware part", "hardware subpart", and "hardware elementary subpart", in accordance with the ISO26262 definition and terms we understood so that there was no other hardware. That axiom was fulfilled. We did not cover non-system level element and system level element, which was the right choice, because an acquired integrated circuit was found to be an element but neither a non-system level element nor a system level element. There are systems and other hardware that constitute integrated circuits, which would violate a covering axiom. Thus, classification is an intricate and precise task and loosening definitions and their use can support the building of the knowledge base from the ontology.

### 5.2.2. Other Challenges/Experiences

- **Training:** Although the definitions appeared rather simple to the expert, practice showed that even for trained personnel, conducting the classifications was challenging. Even after hours of training, usually, an ontologist is often still needed to resolve challenges and ambiguities, which is time-consuming and also costly. The accuracy of the trained personnel may be lower than the possible ideal and needs to be taken into account when calculating overall accuracy. On the other hand, the axioms can be tested if they still hold true for larger amounts of data and if the reasoning can be applied consistently, which also improves the ontology itself.
- **Top-Level:** The upper ontology proved useful. For example, distinguishing functions from processes, which are not only ontologically fundamentally different but also have important practical implications. The beginner may not notice that when they, for example, only use a domain ontology model without top level such as BFO for tagging.
- **Natural Language:** Ambiguities arise from building knowledge graphs from natural language documents. Often, when manually classified, careful revisions are possible and take place, examples are added, other additions such as source links are provided, metadata added, and so forth. In NL documents, sometimes terms are in plural, abbreviations slightly change, and repetitions occur. Furthermore, maybe most importantly, the structure has to be carefully thought about in terms of how to fit in some natural language constructs with the semantic triple or description logic constructs. Sometimes, there is more than one way with different theoretical or practical implications on how to build a knowledge graph.
- **Ontology vs. Knowledge Graph:** Ontologies constitute definitions, formal and informal, hierarchies of taxonomies, and other axioms as well as metadata. They contain few to moderate amounts of classes but they are well-considered. Our initial expectation as well as set up context was to establish the knowledge graph in a way that all necessary axioms would be present to perform reasoning. However, descriptions in, e.g., Wikipedia articles contain the definition only in the beginning and most of the other text only contains the words without explicit structures. Hence, relationships were (1) underrepresented and (2) often present without direct and explicit axioms. It is to be noted that this is not necessarily a limitation of the work and relationships or edges can be constructed using other means or based on referenced or related articles.

Table 10 summarizes the challenges discussed above.

**Table 10.** Summary of challenges.

| Challenges | Descriptions | Proposed/Used Solutions |
|---|---|---|
| Over and under-classification. | Class imbalance. | Resampled to over-sample the minority classes by adding more examples. |
| The adequacy of the current ontology model. | Ontological commitment was very tight and fitting of our ontology in general. | Revise and evaluate your ontological model after applying ML methods and gathering your knowledge base content. |
| Training. | Labeling data required expert intervention. The accuracy of trained personnel may be lower than the expert. | Axioms could still be tested and reasoning could be applied. |
| Top-level. | Trained personnel may not have understood the subtle differences between closely related classes. | Continuously integrate knowledge and advantages of top level ontologies. |
| Natural language. | Ambiguities arise from building knowledge graphs from natural language documents. | Expert guidance and evaluation used. |
| Ontology vs. Knowledge graph. | Ontologies are small, very thoughtful, and highly accurate human build reference models, whereas knowledge graphs contain a significant amount of data and are most often machine-generated. | Combining both realities, building high-quality knowledge graphs based on ontology as reference and scientific approach. |

## 6. Conclusions

In this article, we demonstrated how named entity recognition (NER) can be applied to create knowledge bases in accordance with the ISO26262 definitions in the context of the German GENIAL! Basic Ontology. The application is certainly not limited to this particular domain and can be extended to many more classification and processing tasks, provided enough labeled data are available. We were able to achieve acceptable results and the graph created in the end could in fact be used as an appreciable starting point by a knowledge engineer. This application of NER can help knowledge engineers get a head start on an expert document by looking at the so-called map (Figures 8 and 9) of the entire article in the form of graphs that bring forward a summary of the document in terms of the classes mentioned in it. Furthermore, when using a transformer neural network [48], we were able to generate many more comprehensive sets of graphs from the NER results (see Figure 9). The idea of knowledge-base creation using a proper pipeline of neural networks appears promising. The application of NER in this area of research is certainly worth considering. We believe that this paper is a step in that direction.

Our future work involves expanding our dataset by adding more sentences from expert articles and generating more training and validation data. This will most certainly improve the performance of the model. A better recognition of the class samples would translate into more complex and comprehensive graph generation in the end. We are also currently working on reworking our graph generation method. The simple rule-based algorithm will be replaced by a transformer neural network. This will help in generating more intelligent and elaborate graphs that will assist knowledge engineers in creating knowledge bases from expert documents. We also want to add more tags from our ontology module suite (e.g., "innovation", "disruption", and "mechanical object") and together we expect the results to become substantially more complex and promising. Furthermore, we would like to extend the variety of expert articles to complete automotive roadmaps and datasheets of electric components, including targeting future components. Additionally, we are considering a tight integration loop between an OWL reasoner and the NER results together with human feedback. On the ML side, we further want to migrate

to high-computing processing clusters and integrate more recent methods [34–38] into a combined pipeline to match our computational requirements and resources.

**Author Contributions:** Conceptualization, K.A.R. and F.W.; methodology, F.W.; software, K.A.R. and F.R.; validation, F.W. and F.R.; formal analysis, F.W., K.A.R. and F.R.; investigation, F.W.; resources, F.W. and K.A.R.; data curation, F.R.; writing—original draft preparation, K.A.R. and F.W.; writing—review and editing, F.W.; visualization, K.A.R., F.R. and F.W.; supervision, F.W. and K.A.R.; project administration, K.A.R. and F.W.; funding acquisition, C.G. All authors have read and agreed to the published version of the manuscript.

**Funding:** This work has been supported by the GENIAL! project with funding from the BMBF under grant agreement No 16ES0865K. This work also has received funding from the EU ECSEL Joint Undertaking under grant agreement no. 826452 (project Arrowhead Tools) and from the partners' national funding authorities BMBF under the no. 16ESE0359.

**Institutional Review Board Statement:** Not applicable.

**Informed Consent Statement:** Not applicable.

**Data Availability Statement:** Public repository of our project: https://github.com/khushnood-rafique/Ontology-Learning-Applications-of-Knowledge-Base-Construction, (accessed on 23 December 2022). Public repository of our project with captured state for this paper, including current state of tags, predictions, and code is provided. Repository of knowledge bases and ontologies of the GENIAL! project: https://github.com/wawrzik/GENIALOntologies, (accessed on 23 December 2022). Subfolder containing the GENIAL! Basic ontology: https://github.com/wawrzik/GENIALOntologies/tree/master/OntologyModuleSuite/GENIAL!BasicOntology, (accessed on 23 December 2022).

**Acknowledgments:** We thank Vishwanath Sathyanarayana for contributing to the graph relationship generation.

**Conflicts of Interest:** The authors declare no conflict of interest. The funders had no role in the design of the study; in the collection, analyses, or interpretation of data; in the writing of the manuscript; or in the decision to publish the results.

## Abbreviations

The following abbreviations are used in this manuscript:

| | |
|---|---|
| OWL | Ontology Web Language |
| KG | Knowledge Graph |
| GENIAL | Common Electronics Roadmap for Innovations along the Automotive Value Chain |
| SysMD | System MarkDown |
| Bi-LSTM | Bi-directional Long Short Term Mermory |
| GBO | GENIAL! Basic Ontology |
| BFO | Basic Formal Ontology |

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
