# Peer review of "Ontology Learning Applications of Knowledge Base Construction for Microelectronic Systems Information"

_information, doi:10.3390/info14030176_

Round 1
Reviewer 1 Report
The study aims to solve the knowledge base construction (presented in the form of graphs) from the information available in textual format task using AI-based NLP approaches.
The task is relevant for many domains and knowledge bases presented in different forms like graphs, ontologies, etc.
The approach is based on its own dataset based on vocabulary ISO 26262. It uses LSTM to make more fine-grained distinctions of definitions of classes when extracting the classes and relations.
The dataset was formed with Wikipedia articles with labeled tags.
Bi-LSTM was used to train the model. The classes extraction task was considered as a Named entity recognition task. The second step involved creating a graph using the identified elements and relations detected using context.
The approach applied for the microelectronics domain and showed F1 measure value of 0.64 which is good enough to use the approach to help the knowledge engineer in the domain knowledge base construction.
Author Response
We thank you for the review.
Reviewer 2 Report
This paper presents and tests a novel process to generate part of a knowledge base from natural language statements of technical articles in the microelectronics domain.
This paper has some positive aspects: despite existing research works on the construction of semantic graphs from NL statements since the 90’s (see for example Alecsi in the CAISE conference proceedings), this topic is still valid and interesting. The fact that it deals specifically with microelectronics Information Systems adds to its novelty. The proposed process combines modern & powerful techniques ( i.e. neural network model) with less recent ones (i.e. named entity recognition). The knowledge base obtained from the test data shows that the process « predicted a good number of class samples, correctly ».
The paper has also a large number of drawbacks that considerably limit its contribution and value:
- first, the paper is difficult to follow (despite a very good English writing), cumbersome in many places, making not easy the understanding of the research proposal and its exact contribution.
- The main text (the 15 first pages) is very condensed, remains too general & superficial in parts without concrete examples to facilitate understanding whereas the 6 pages of appendix A and appendix B are useless. These pages would be better used in the main part of the paper.
- Whereas the abstract presents NL processing as the central aspect of the contribution of the paper , this aspect is scarcely dealt with in the paper and the reader has to reach page 12 (among 15) to find some examples of NL statements and their treatment.
- Frankly, Figure1 which is supposed to sum up the proposed approach is incomprehensible. The only sentence provided as an explanation ‘In the first phase, we tag text articles according to our methodology and definitions and in a second step we generate the relationships between the classes’ is insufficient.
-The focus is on the Ontology and the dataset. Why not? But this requires to change the presentation of the paper and its contribution.
- The test is insufficiently developed. A full research paper which proposes a new approach must contain a serious evaluation. This is not at all the case. A simple statement from authors (see above) without any proof of what is stated, is absolutely insufficient.
- last but not least, this paper is a project report like the many ones which we have to produce in an EU project or nationally financed project. It says what has been done, in a descriptive mode, without no focus on WHY it is done that way which is key for a research paper. It does not respect the typical paradigm for presenting a new research: state the problem and give evidences for it, proposes hypotheses to solve the problem, develop a solution to confirm/deny hypotheses, evaluate the solution including comparisons with existing ones.
Overall, the technical material of this paper is certainly sufficient to be explained in a scientific manner and journal publishable mode but this is not the case in this version of the paper.
Author Response
The paper has also a large number of drawbacks that considerably limit its contribution and value:
- first, the paper is difficult to follow (despite a very good English writing), cumbersome in many places, making not easy the understanding of the research proposal and its exact contribution.
The exact contribution lies at the intersection of a new dataset with a new axiomatized population approach, using NL. Meaning the contribution is more on the ontology side but not independent of the NL addition.
- The main text (the 15 first pages) is very condensed, remains too general & superficial in parts without concrete examples to facilitate understanding whereas the 6 pages of appendix A and appendix B are useless. These pages would be better used in the main part of the paper.
We redistributed the appendix and integrated it in the main part, on page 5 (approach), page 16 (Discussion) where we deem them to fit best
The main illustrative example is given in figure 2, in order to represent the population of the graph.
- Whereas the abstract presents NL processing as the central aspect of the contribution of the paper , this aspect is scarcely dealt with in the paper and the reader has to reach page 12 (among 15) to find some examples of NL statements and their treatment.
I am not sure how to change the abstract to make it clearer.
- Frankly, Figure1 which is supposed to sum up the proposed approach is incomprehensible. The only sentence provided as an explanation ‘In the first phase, we tag text articles according to our methodology and definitions and in a second step we generate the relationships between the classes’ is insufficient.
We acknowledge figure 1 needed to be improved. We updated the figure accordingly
-The focus is on the Ontology and the dataset. Why not? But this requires to change the presentation of the paper and its contribution.
I felt to keep the ontology part just big enough to understand the use, not to introduce the whole ontology in detail (other paper). The intention was to create a balance of around 50/50. See previous answer.
- The test is insufficiently developed. A full research paper which proposes a new approach must contain a serious evaluation. This is not at all the case. A simple statement from authors (see above) without any proof of what is stated, is absolutely insufficient.
We see the evaluation as ongoing and as a grande challenge of that field of research in general, which is just starting out, needs separate focus and will be an accompaniment for some time to come.
- last but not least, this paper is a project report like the many ones which we have to produce in an EU project or nationally financed project. It says what has been done, in a descriptive mode, without no focus on WHY it is done that way which is key for a research paper. It does not respect the typical paradigm for presenting a new research: state the problem and give evidences for it, proposes hypotheses to solve the problem, develop a solution to confirm/deny hypotheses, evaluate the solution including comparisons with existing ones.
As mentioned in the introduction, that approach only now becomes feasible, because of new higly parallelised reasoners which scale on high computing clusters. What we do was not computable and still is not with industry standard reasoners like hermit, pellet and fact.
Reviewer 3 Report
1. In table 1, I am not satisfied with the assumption of Ontology as CWA, authors should justify it with other references where it is recommended.
2. Most of the figures and tables are taken from other sources, author should include their own findings too.
3. Authors have proposed a ontology GBO but not discussed about the evaluation of this ontology although they have discussed about the evaluation metrics of proposed model.
4. Authors are suggested to improve the paper by describing more about ontology learning as it is only discussed in the title of the paper.
Author Response
- In table 1, I am not satisfied with the assumption of Ontology as CWA, authors should justify it with other references where it is recommended.
I agree and I am not either. Ontologies, in particular OWL-DL 2 is built upon the open world assumption, which allows to construct and reason with incomplete knowledge and requires explicit negation or disjunction in order to create inconsistencies. Further it allows not to conclude that something is untrue or not the case if absent. As I referenced the table, I pointed out some of the errors by the authors made with the table but retained the table as to adhere to proper citing and referencing. I addressed that point in line 94. Appropriate references have been added that support the points made.
In the table, we assume the authors made a mistake by attributing the closed world assumption to ontologies in table row 1 (as substantiated in e.g. \cite{ALLEMANG2011249, 10.1007/978-3-642-25832-9_80}) and mistakenly swapped the easy data integration of ontologies with the harder to integrate knowledge graphs (row 11, referred to in e.g. \cite{Gusenkov_2019, Calvanese2018}).
2. Most of the figures and tables are taken from other sources, author should include their own findings too.
The tables and figures are all original created from the scratch. The references are now removed. Table 1 is the only taken “figure”/table from other sources than the authors.
3. Authors have proposed a ontology GBO but not discussed about the evaluation of this ontology although they have discussed about the evaluation metrics of proposed model.
I added a comprehensive paragraph in line 195 addressing a more general evaluation of just the GBO ontology.
We evaluated the ontology from various angles, including application and quality of soundness of the model. We classify it as an intermediate middle level, expressive, small (80 atomic classes), core ontology. It is not just for tagging and data aggregation, but use in the application itself. Several domain experts from Bosch, Infineon, Hella, etc. have given feedback and revisions. The outcome was that the domain was well represented, both in detail and correctness. Sometimes users had preferences to additions, like including the ASIL safety level (Bosch), which was added to extensions in other parts of a module suite. It further showed, that the ontology made even more clear the meaning of the words, as it was making implicit assumptions of the textual ISO26262 standard explicit \cite{Wawrzik2022}. Another result of the evaluation was, that the model was still hard to understand and use for non experts, and we created a more abstract simplified and intuitive vocabulary as a consequence and complement \cite{SysMD}. An evaluation in terms of speed took place as well \cite{DBLP:conf/semweb/WawrzikL21}. The GBO ontology took a reasoning time between 2-15 minutes depending on the reasoner and machine for a hardware software domain example. The evaluation yielded necessary improvements in approaches like modularisation and allowed for the creation of this work as a next logical step.
4. Authors are suggested to improve the paper by describing more about ontology learning as it is only discussed in the title of the paper.
A small paragraph has been added to explain ontology learning at the beginning of the introduction with appropriate references.
Ontology learning is a specific part of automatic knoweldge base construction which involves machine learning and TBox axioms. Tbox axioms, or terminological axioms describe information regarding the description of ideas and concepts rather than actual tangible and concrete information. It is distinguished from ontology population, which addresses the concrete instances and triples of the construction process, also called ABox or assertions box. Ontology learning includes, but is not limited to axioms like existential restrictions, universal restrictions, cardinality restrictions, disjunct classes, building taxonomic and subclass of hierarchies, and more. \cite{WATROBSKI20203356, KHADIR2021100339} gives an overview and state of the art summary.
Reviewer 4 Report
A brief summary
This paper is about building a knowledge base for microelectronics.
They share the knowledge base and have a tutorial.
Specific comments:
This paper has a clear motivation and decent experimental results.
I am leaning to give a "minor revisions" based on my current knowledge and understanding of the paper. But I will be willing to revisit the decision after get feedback from the author(s).
In particular, I would be glad if the author could clarify the questions below.
*The process of building the knowledge base looks a bit messy and needs a figure to show the building process.
*Figure 2 has too much information content, can it be streamlined?
*With respect to your work in building the knowledge base, why use Bi-LSTM models for prediction?
Now SOTA models should be Autoformer (NeurIPS'21), Informer (AAAI'21 Best paper), LogTrans (NeurIPS'2019), Reformer (ICLR 2020), FEDformer (ICML 2022) these methods. You can read this paper to understand the progress of sequence prediction models:
[Survey] Qingsong Wen, Tian Zhou, Chaoli Zhang, Weiqi Chen, Ziqing Ma, Junchi Yan, Liang Sun, "Transformers in Time Series: A Survey," arXiv preprint arXiv:2202.07125 (2022).
*The Section 3.4 should indicate the details of the person who manipulated the dataset, e.g., age, gender, job, linguistic ability, etc.
*How are the parameters of the Bi-LSTM model set? Were they determined after comparison and verification?
*The results of the sys data in Table 5 are relatively lower than the other results, is the reason the same as the low results of the func data?
*Is the validation process for the knowledge base all black box?
Is there any overlap between evaluators and knowledge base builders? If there is overlap, how can the impact on the results be avoided?
*In the General Challenges Section, the authors discuss a lot of issues, which is good. But a table or graph is needed for a summary so that it looks logical.
Author Response
The process of building the knowledge base looks a bit messy and needs a figure to show the building process
Figure was changed.
With respect to your work in building the knowledge base, why use Bi-LSTM models for prediction? Now SOTA models should be Autoformer (NeurIPS'21), Informer (AAAI'21 Best paper), LogTrans (NeurIPS'2019), Reformer (ICLR 2020), FEDformer (ICML 2022) these methods. You can read this paper to understand the progress of sequence prediction models: [Survey] Qingsong Wen, Tian Zhou, Chaoli Zhang, Weiqi Chen, Ziqing Ma, Junchi Yan, Liang Sun, "Transformers in Time Series: A Survey," arXiv preprint arXiv:2202.07125 (2022)
We acknowledge the existence of more advanced and recent approaches and added our rationale of choice in line 129. We added the references for completeness:
We acknowledge that transformers could be more potent for NER tasks now. Since we had limited hardware resources and the scale of our project was on the smaller side, we tried to keep it simple and used a Bi-LSTM for our classification task. Our Bi-LSTM model was less computationally expensive which makes them more compatible for real-time or resource-constrained applications. Transformers on the other hand are more computationally expensive on longer sequences.
The Section 3.4 should indicate the details of the person who manipulated the dataset, e.g., age, gender, job, linguistic ability, etc
We added the information at line 233 in the report:
The assistants were master students of computer science. One assistant was a male in his late twenties while the other was a female in her mid-twenties. Both could use the English language fluently, though as a second language and accurately in a professional setting.
How are the parameters of the Bi-LSTM model set? Were they determined after comparison and verification?
We added the answer to your question in the report in line 273:
We trained the lstm model with index numbers since it cannot recognize text. Every word and tag was assigned a unique index number to simplify the process. Hence the following two new rows Word\_to\_index and Tag\_to\_index) were generated in the dataset:
Word\_to\_index: Assign an index number to each word. These numbers were used during training to identify the words.
Tag\_to\_index: Assign an index number to each tag. These numbers were used during training to identify the tags.
We trained for 50 epochs as that gave us the best training loss curve where over- and under-fitting were best avoided. The training loss curve remained below the validation loss curve. We used ‘categorical crossentropy’, which we found most appropriate for a multi-class classification task. Other hyper-parameters such as the LSTM units, dropouts, were adjusted with multiple passes of experimentations and comparisons. As a result, we reached the validation accuracy was 94\%.
The results of the sys data in Table 5 are relatively lower than the other results, is the reason the same as the low results of the func data?
Yes: lack of sufficient training data for the particular classes.
Is the validation process for the knowledge base all black box?
I added an outline of validation at line 386:
\subsection{Validation}
Our validation process is twofold:
\subsubsection{Manually} After the graph is generated, we mostly check for semantic accuracy, coherency and consistency, manually and update the dataset. Other criteria like timeliness, understandability and completeness are out of scope. Two validation examples are given
\begin{itemize}
\item the transformer uses a relationship “uses” between parts and functions, we classify it as subclass of “implements” in order to adhere to our schema and retain the other information as well (Coherency)
\item We recognize if a property was falsely classified as function, and readjust the dataset and graph (Semantic Accuracy)
\end{itemize}
\subsubsection{Automatically}
We apply the reasoner and create an inconsistent knowledge graph and dataset as well as a consistent one automatically which simplifies manual work
Is there any overlap between evaluators and knowledge base builders? If there is overlap, how can the impact on the results be avoided?
I do not know.
In the General Challenges Section, the authors discuss a lot of issues, which is good. But a table or graph is needed for a summary so that it looks logical.
Table 10 was added. It gives an overview
Round 2
Reviewer 2 Report
Thanks for taking comments into account. The paper is publishable in its current form.